# Istaroxime for Patients with Acute Heart Failure: A Systematic Review and Meta-Analysis of Randomized Controlled Trials

**DOI:** 10.3390/diseases11040183

**Published:** 2023-12-17

**Authors:** Mohamed Abuelazm, Shafaqat Ali, Majd M. AlBarakat, Abdelrahman Mahmoud, Mohammad Tanashat, Husam Abu Suilik, Basel Abdelazeem, James Robert Brašić

**Affiliations:** 1Faculty of Medicine, Tanta University, Tanta 31527, Egypt; dr.mabuelazm@gmail.com; 2Department of Internal Medicine, Louisiana State University Health Shreveport, Shreveport, LA 71103, USA; shafaqat231@gmail.com; 3Faculty of Medicine, Jordan University of Science and Technology, Irbid 22110, Jordan; mmalbarakat20@med.just.edu.jo; 4Faculty of Medicine, Minia University, Minia 61519, Egypt; abdelrahman.mahmoud20211997@gmail.com; 5Faculty of Medicine, Yarmouk University, Irbid 21163, Jordan; mohammad_tanashat@hotmail.com; 6Faculty of Medicine, Hashemite University, Zarqa 13133, Jordan; husamabusuilik.md@gmail.com; 7Division of Cardiology, Department of Medicine, West Virginia University School of Medicine, Morgantown, WV 26506, USA; baselelramly@gmail.com; 8Section of High-Resolution Brain Positron Emission Tomography Imaging, Division of Nuclear Medicine and Molecular Imaging, The Russell H. Morgan Department of Radiology and Radiological Science, The Johns Hopkins University School of Medicine, Baltimore, MD 21287, USA; 9Department of Psychiatry, New York City Health and Hospitals/Bellevue, New York, NY 10016, USA; 10Department of Psychiatry, New York University Grossman School of Medicine, New York University Langone Health, New York, NY 10016, USA

**Keywords:** atrial fibrillation, blood pressure, clinical trial, confidence interval, forest plot, inotropic, mean difference, probability, outcome, shock

## Abstract

Istaroxime, an intravenous inotropic agent with a dual mechanism—increasing both cardiomyocyte contractility and relaxation—is a novel treatment for acute heart failure (AHF), the leading cause of morbidity and mortality in heart failure. We conducted a systematic review and meta-analysis that synthesized randomized controlled trials (RCTs), which were retrieved by systematically searching PubMed, Web of Science, SCOPUS, and Cochrane until 24 April 2023. We used a fixed-effect or random-effect model—according to heterogeneity—to pool dichotomous data using the risk ratio (RR) and continuous data using the mean difference (MD), with a 95% confidence interval (CI). We included three RCTs with a total of 300 patients. Istaroxime was significantly associated with an increased left ventricular ejection fraction (mL) (MD: 1.06, 95% CI: 0.29, 1.82; *p* = 0.007), stroke volume index (MD: 3.04, 95% CI: 2.41, 3.67; *p* = 0.00001), and cardiac index (L/min/m^2^) (MD: 0.18, 95% CI: 0.11, 025; *p* = 0.00001). Also, istaroxime was significantly associated with a decreased E/A ratio (MD: −0.39, 95% CI: −0.58, −0.19; *p* = 0.0001) and pulmonary artery systolic pressure (mmHg) (MD: 2.30, 95% CI: 3.20, 1.40; *p* = 0.00001). Istaroxime was significantly associated with increased systolic blood pressure (mmHg) (MD: 5.32, 95% CI: 2.28, 8.37; *p* = 0.0006) and decreased heart rate (bpm) (MD: −3.05, 95% CI: −5.27, −0.82; *p* = 0.007). Since istaroxime improved hemodynamic and echocardiographic parameters, it constitutes a promising strategy for AHF management. However, the current literature is limited to a small number of RCTs, warranting further large-scale phase III trials before clinical endorsement.

## 1. Introduction

Heart failure (HF) is a chronic and progressive disease estimated to have a prevalence of over 26 million worldwide, and it is on the rise, with an overwhelming global healthcare-related burden [1,2,3]. The exacerbation of acute heart failure (AHF) is the leading cause of morbidity and mortality associated with HF. The natural history of AHF has largely remained unchanged in the last decade [4,5]. Patients with low systolic blood pressure (SBP) or overt cardiogenic shock (CS) continue to be at the highest risk of poor outcomes [6,7,8]. The Society of Cardiovascular Angiography and Interventions (SCAI) has recently proposed the classification of CS into five stages (A to E) to characterize patients for better-targeted treatment [9]. Pre-cardiogenic shock (Pre-CS) refers to the period of rapid hemodynamic deterioration that precedes overt CS with hypotension, inflammatory response, and end-organ damage [10].

The current management strategies for AHF with CS are inotropic agents (e.g., dobutamine, milrinone, and enoximone) and vasoactive agents (e.g., norepinephrine). Unfortunately, despite the transient improvement in hemodynamic status with these agents, they have not shown survival benefits [11,12]. In addition, all inotropic drugs recommended for patients with AHF activate adrenergic signaling to some extent, and extended use of these drugs can be potentially harmful [12]. Istaroxime is a novel intravenous inotropic agent with a dual mechanism that inhibits Na^+^/K^+^ adenosine triphosphatase activity while activating sarcoplasmic reticulum Ca^+2^ adenosine triphosphatase isoform 2a (SERCA2a) [13,14,15]. This dual mechanism causes an increase in intracellular calcium and promotes calcium reuptake in the sarcoplasmic reticulum, resulting in both increased cardiomyocyte contractility (during systole) and relaxation (during diastole) [16,17,18]. This could offset the risk of arrhythmias associated with traditional inotropes. These effects support a potential therapeutic role for istaroxime in treating AHF, and recent clinical trials have studied its use in these patients [19,20,21,22,23].

To clarify the safety and efficacy of istaroxime for AHF, a systematic review and meta-analysis determined that istaroxime is effective in (A) increasing the left ventricular ejection fraction (LVEF), cardiac index, and systolic blood pressure (SBP) and (B) reducing the E/A ratio, indicating improved left ventricular function, left ventricular end-diastolic volume (LVEDV), and left ventricular end-systolic volume (LVESV) [24]. We sought to conduct a systematic review and meta-analysis utilizing more extensive criteria to extract more data [25,26] to further investigate the safety and efficacy of istaroxime in AHF.

## 2. Materials and Methods

### 2.1. Protocol Registration

We conducted this review by following the Preferred Reporting Items for Systematic Reviews and Meta-Analyses (PRISMA) guidelines [27] and the Cochrane Handbook for Systematic Reviews and Meta-Analyses [28]. We pre-registered and published the protocol for this study on PROSPERO (CRD42023424614).

### 2.2. Data Sources and Search Strategy

M.A. and B.A. conducted an electronic search through five databases: PubMed (MEDLINE), Scopus, Cochrane Central Register of Controlled Trials (CENTRAL), Web of Science (WoS), and EMBASE were searched until 24 April 2023. We also conducted an updated search just before starting the analysis to include any recently published articles after the original search. We modified the search terms and keywords for each database (Appendix A).

### 2.3. Eligibility Criteria and Study Selection 

We included studies that followed the following PICO criteria: population (patients with AHF); intervention (istaroxime irrespective of the dosage); comparison (placebo); outcomes (echocardiographic parameters (left ventricle (LV) end-diastolic volume, LV end-systolic volume, LV ejection fraction, stroke volume index, cardiac index, E/A ratio, E/e′ ratio, inferior vena cava (IVC) diameter, and pulmonary artery systolic pressure), hemodynamic parameters (systolic blood pressure (SBP), mean arterial pressure (MAP), and heart rate (HR)), clinical outcomes (NT-proBNP change, length of hospital stay, worsening HF, and hospital readmission), and safety outcomes (the incidence of any adverse events including non-serious adverse events (nausea, vomiting, and injection site pain) and serious adverse events (cardiac arrest, ventricular tachycardia, pneumonia, acute renal failure, and newly diagnosed coronary artery disease)); and study design (randomized clinical trials (RCTs)). 

We excluded studies that met any of the following criteria: (1) non-original studies (e.g., reviews, book chapters, correspondence, letters to editors, commentaries, press articles, and guidelines), (2) any study designs other than RCTs, (3) studies with overlapping datasets or duplications, (4) studies with a sample size of fewer than ten participants, (5) studies that did not report on istaroxime or lacked a primary outcome, (6) in vitro experiments and non-human studies, and (7) studies that were not conducted in the English language.

### 2.4. Study Selection

We utilized the online Covidence tool [29] to conduct the review process. After removing any duplicate records, M.M.A. and M.T. independently reviewed the retrieved records. During the full-text screening, M.M.A. and M.T. checked the full texts of the records that met the original eligibility requirements. Any disagreements that arose were resolved through discussion and agreement with a senior author.

### 2.5. Data Extraction

Four reviewers (A.M., H.A., M.A., and M.T.) independently used a well-designed online extraction form to extract the upcoming data. The first part included the summary characteristics of the included studies (name of first author, year of publication, country, name of journal, and study design). The second part included the baseline information of the participants (sample size, age, gender, follow-up period, SBP, HR, LV end-diastolic volume, LV end-systolic volume, LV ejection fraction, stroke volume index, cardiac index (L/min/m^2^), E/A ratio, and E/e′ ratio). Finally, the third part included outcome data. The data extraction process was carried out by two reviewers (M.M.A. and M.T.). Any disagreements that arose were resolved through discussion and agreement with a senior author.

### 2.6. Risk of Bias and Certainty of Evidence

Two investigators (H.A. and M.T.) independently appraised the property of the studies using the Cochrane ROB2 tool [30]. Any disagreements that arose were resolved through discussion and agreement with a senior author. To evaluate the quality of the evidence, two reviewers (M.A. and B.A.) used the Grading of Recommendations Assessment, Development, and Evaluation (GRADE) guidelines [25,26]. Any differences were settled via consensus.

### 2.7. Statistical Analysis

The RevMan v5.3 software [31] was the statistical analysis application of choice for conducting the analysis. The mean difference (MD) was used for continuous results, and the risk ratio (RR) was used for dichotomous outcomes. Both were calculated with a 95% confidence interval (CI) using a fixed-effect model. If the heterogeneity was high, we used a random-effect model. The heterogeneity was evaluated using the Chi-square and I-square tests, respectively, according to the Cochrane Handbook (chapter nine) [26].

## 3. Results

### 3.1. Search Results and Study Selection

The search process yielded 216 studies that were screened and evaluated for their titles and abstracts. After duplicates (106) and irrelevant studies (105) were excluded, five full-text articles were screened. Finally, three studies were included in the qualitative and quantitative synthesis (Figure 1).

### 3.2. Characteristics of the Included Studies

We included three RCTs [19,20,21], with a total of 300 participants. Details of the included RCTs and the baseline characteristics of the participants are reported in Table 1 and Table 2, respectively.

### 3.3. Risk of Bias and Certainty of Evidence

All of the included trials showed a low risk of bias across the assessed domains (Figure 2, Appendix A). Also, the certainty of evidence was outlined in a GRADE evidence profile (Table 3).

### 3.4. Echocardiographic Parameters

Istaroxime was significantly associated with an increased LV ejection fraction (mL) (MD: 1.06; 95% CI: 0.29, 1.82; *p* = 0.007), stroke volume index (MD: 3.04; 95% CI: 2.41, 3.67; *p* = 0.00001), and cardiac index (L/min/m^2^) (MD: 0.18; 95% CI: 0.11, 0.25; *p* = 0.00001). Also, istaroxime was significantly associated with a decreased E/A ratio (MD: −0.39; 95% CI: −0.58, −0.19; *p* = 0.0001) and pulmonary artery systolic pressure (MD: −2.30; 95% CI: −3.20, −1.40; *p* = 0.00001). However, there was no difference between istaroxime and the placebo regarding the LV end-diastolic volume (MD: −4.69; 95% CI: 12.85, 3.48; *p* = 0.26), LV end-systolic volume (MD: −5.40; 95% CI: −12.05, 1.25; *p* = 0.11), E/e′ ratio (MD: −1.04; 95% CI: −4.15, 2.07; *p* = 0.51), and IVC diameter (mm) (MD: −1.82; 95% CI: −3.74, 0.15; *p* = 0.06) (Figure 3).

Our results were homogenous for the LV ejection fraction (*p* = 0.22, I^2^ = 32%), stroke volume index (*p* = 0.87, I^2^ = 0%), cardiac index (*p* = 0.30, I^2^ = 18%), E/A ratio (*p* = 0.24, I^2^ = 29%), and pulmonary artery systolic pressure (*p* = 0.49, I^2^ = 0%). However, our results were heterogenous for the LV end-diastolic volume (*p* = 0.26, I^2^ = 92%), LV end-systolic volume (*p* = 0.00001, I^2^ = 91%), E/e′ ratio (*p* = 0.00001, I^2^ = 93%), and IVC diameter (*p* = 0.0007, I^2^ = 86%). We performed a sensitivity analysis, and the heterogeneity was best resolved by excluding Metra et al. (2022) [19] (*p* = 0.81, I^2^ = 29%; *p* = 0.84, I^2^ = 32%; *p* = 0.54, I^2^ = 0%; *p* = 0.38, I^2^ = 0%) (Appendix A).

### 3.5. Hemodynamic Parameters

Istaroxime was significantly associated with increased SBP (mmHg) (MD: 5.32; 95% CI: 2.28, 8.37; *p* = 0.0006) and decreased HR (MD: −3.05; 95% CI: −5.27, −0.82; *p* = 0.007). However, there was no difference between istaroxime and the placebo regarding MAP (mmHg) (MD: 2.44; 95% CI: −0.17, 5.05; *p* = 0.06) (Figure 4). Our results were homogenous for SBP (*p* = 0.70, I^2^ = 0%), HR (*p* = 0.54, I^2^ = 0%), and MAP (*p* = 0.48, I^2^ = 0%). 

### 3.6. Clinical Parameters

Istaroxime was significantly associated with an increased NT-proBNP level (MD: 808.28; 95% CI: 523.98, 1092.58; *p* = 0.00001) (Figure 5A). However, there was no difference between istaroxime and the placebo regarding the length of stay in the hospital (days) (MD: −0.23; 95% CI: −1.66, 1.21; *p* = 0.76) (Figure 5A), worsening HF (RR: 1.88 (95% CI: 0.49, 7.20; *p* = 0.35), or hospital readmission (RR: 1.09 (95% CI: 0.25, 4.71; *p* = 0.91) (Figure 5B).

Our results were homogenous for the NT-proBNP level (*p* = 0.88, I^2^ = 0%), length of stay in the hospital (*p* = 0.17, I^2^ = 43%), worsening HF (*p* = 0.36, I^2^ = 3%), and hospital readmission (*p* = 0.31, I^2^ = 16%). 

### 3.7. Safety Outcomes

Istaroxime was significantly associated with an increased rate of incidence of any adverse events (RR: 1.26; 95% CI: 1.05, 1.51; *p* = 0.01). However, there were no differences between istaroxime and the placebo regarding any serious adverse events (RR: 1.34; 95% CI: 0.58, 3.11; *p* = 0.49) (Appendix A). Our results were homogenous for the incidence of any adverse events (*p* = 0.34, I^2^ = 0%) and any serious adverse events (*p* = 0.52, I^2^ = 0%).

## 4. Discussion

We report several noteworthy findings in this meta-analysis of RCTs evaluating the efficacy and safety of istaroxime in AHF patients. First, istaroxime was significantly associated with an increased LV ejection fraction, stroke volume, and cardiac index, as well as a decreased E/A ratio and pulmonary artery systolic pressure. However, there was no significant difference between LV end-diastolic volume and LV end-systolic volume compared to the placebo. Second, istaroxime therapy was significantly associated with increased SBP and decreased HR, but it could not achieve statistical significance regarding a higher MAP than that of the placebo group. Third, despite a significant increase in the NT-proBNP level, there was no difference in other important clinical outcomes, including the length of stay, hospital readmission rate, and worsening of HF. Finally, there was no statistically significant difference in the incidence of serious adverse events compared to the placebo. However, it increased the rate of other non-serious adverse events, such as infusion-related side effects.

The mechanism of action of istaroxime can explain the echocardiographic findings observed in our study. Understanding the pathophysiology of excitation–contraction (EC) coupling is vital when interpreting the effects of any inotropic therapy for HF. In patients with AHF, the inability of the heart to eject sufficient blood for peripheral tissue perfusion is caused by defects in EC coupling in cardiac myocytes [32]. Istaroxime works through a dual mechanism that inhibits Na^+^/K^+^ adenosine triphosphatase activity while activating sarcoplasmic reticulum Ca^+2^ adenosine triphosphatase isoform 2a (SERCA2a) [6,13,14]. The inhibition of Na^+^/K^+^ adenosine triphosphatase increases intracellular sodium levels, affecting Ca^+2^ homeostasis through Na^+^/Ca^+2^ exchange (NCX) and resulting in increased levels of intracellular Ca^+2^. Intracellular Ca^+2^ then binds to troponin C, facilitating actin–myosin interaction, which induces the contraction of the cardiac myocytes. The role of intracellular Ca^+2^ in homeostasis for cardiomyocyte contractility has been extensively investigated [33,34].

To clarify, during diastole, Ca^+2^ diffuses away from troponin C, initiating relaxation. The Ca^+2^ that was released from the sarcoplasmic reticulum is taken back up by the SERCA2a, whereas the amount of Ca^+2^ that entered the cell via L-type calcium channels is exported by the NCX [35]. However, in a failing heart, the increased cytosolic concentration of Ca^+2^—even during diastole due to impaired Ca^+2^ expulsion via the forward mechanism of NCX—can result in slower relaxation [35]. This can be problematic, especially with an elevated HR, when the diastolic period is already shortened, leading to impaired filling and, subsequently, a progressive decline in cardiac output. Theoretically, through a dual mechanism, istaroxime can offset this effect by improving contractility during systole and relaxation during diastole, leading to an improved stroke volume, LV ejection fraction, and cardiac index, as observed in our analysis. Similarly, we noticed a trend toward decreased end-diastolic and systolic volumes, which did not achieve statistical significance.

The current literature is limited in in its comparison of istaroxime with classic inotropic agents (digoxin, dobutamine) and Ca^+2^ sensitizers (levosimendan). Digitalis-derived cardiotropic glycosides, e.g., digoxin, are the oldest inotropic drugs, and they may be considered in symptomatic HF with a reduced ejection fraction (HFrEF) despite being at the maximum tolerated dose of guideline-directed medical therapy [4,35,36]. In a trial by the Digitalis Investigation Group (DIG), digoxin did not improve all-cause mortality but reduced hospitalization due to HF [35]. However, this trial was conducted before the advent of the current guideline-directed medical therapy for HFrEF. The current American College of Cardiology/American Heart Association (ACC/AHA) and European Society of Cardiology (ESC) guidelines recommend against the routine use of inotropes, except in certain high-risk patient populations with AHF, e.g., patients with signs of cardiogenic shock, SBP < 90 mmHg refractory to fluid challenge, or persistent symptomatic hypoperfusion [4,37]. These guidelines were based on studies evaluating the safety and efficacy of inotropes (dobutamine, dopamine, and phosphodiesterase inhibitors), and neutral effects were reported in the short term, while long-term use was associated with adverse outcomes in patients hospitalized for AHF [38,39,40].

There are no extant data for directly comparing istaroxime with pimobendan and levosimendan, which are calcium sensitizers that increase cardiac myocyte contractility. Nevertheless, drugs in this class may have variable mechanisms. Still, the endpoint is an increased affinity of troponin C to binding Ca^+2^, resulting in an increased force of contraction during systole for any given amount of intracellular Ca^+2^ in cardiac myocytes [41,42]. On the other hand, this also impairs diastolic relaxation due to the slowing of calcium reuptake. However, istaroxime can improve diastolic relaxation by stimulating calcium reuptake. Moreover, the increase in myofilament Ca^+2^ sensitization has been reported to be arrhythmogenic, which is another concern with the use of calcium sensitizers [43].

In this context, it is essential to notice that in the SURVIVE trial, patients on levosimendan with β-blocker pre-treatment had improved short-term survival compared to those taking dobutamine [44]. Another study reported the potential benefit of levosimendan in patients with ischemic heart disease [45]. Conversely, in septic shock patients co-treated with catecholamines, levosimendan was associated with higher rates of supraventricular tachycardia [46]. These observed findings point towards the possibility of adrenergic properties of calcium sensitizers similar to those of classic inotropes; hence, their role in AHF management is not yet fully established [4,12,36]. In our study, istaroxime was associated with decreased HR without significant differences in serious adverse events compared to the control group. This can be clinically important, as the role of beta blockers in acute HF for achieving the desired chronotropic effect is limited due to their negative inotropic effect. Landiolol is an ultra-short-acting cardioselective B1-blocker that has been used for tachyarrhythmias in patients with LV dysfunction [47,48,49,50]. However, there are very limited data to establish its safety in acute HF patients. Hence, future studies should consider a direct comparison of istaroxime with calcium sensitizers and digitalis to evaluate the efficacy and safety of these drugs in comparison with each other.

Finally, our study highlighted a significant increase in NT-proBNP with istaroxime. Although NT-ProBNP is widely used as a diagnostic marker for HF, its prognostic value is still uncertain. It is not well known whether elevated NT-proBNP can be associated with overall poor outcomes, as its levels can be affected by multiple factors, including advanced age, renal insufficiency, and arrhythmias [51,52]. Furthermore, we did not observe any significant differences in other clinically important outcomes, such as the length of hospital stays, worsening HF, hospital readmission rates, or safety outcomes such as serious side effects. On theoretical grounds, treatments that restore the defects of cytosolic Ca^+2^ handling in HF without activating adrenergic signaling can be promising, since they avoid diastolic dysfunction and the long-term adverse consequences of adrenergic activation. However, it is vital to highlight that any therapy (including istaroxime) targeting EC coupling and mitochondrial energetics in cardiomyocytes can potentially induce maladaptive cardiac remodeling and apoptosis through a transition in mitochondrial permeability [53].

### Strengths and Limitations

To the best of our knowledge, this is the most extensive meta-analysis to investigate the efficacy and safety of istaroxime for AHF management, constituting gold-standard evidence, with an extensive assessment of the certainty of the evidence by using the GRADE approach [25,26]. However, the following limitations of our study should be considered before interpreting our results. The studies included in our meta-analysis had a relatively younger patient population with a mean age ranging between 50 and 65 years. Patients with extreme heart rates were excluded from the trials. For example, Shah et al. included only patients with HRs from 60 to 110 beats per minute (bpm) [21], and Carubelli et al. excluded patients with resting HRs of <50 bpm or >120 bpm [20].

The HORIZON-HF trial [21] lacked a core laboratory for the review and quantification of echocardiographic images, which could have potentially resulted in discrepancies in the echocardiographic readings of the study. Also, maneuvers such as preload reduction were not performed for the pressure–volume analyses, which could have impaired their overall accuracy. Furthermore, the pulse-wave tissue Doppler imaging used in this trial had limitations, such as the angle dependence of the Doppler beam, which can be important, especially in patients with dilated cardiomyopathy. Similarly, Carubelli et al. [20] had enrollment discrepancies between the study population and the selection of the two cohorts, with one cohort enrolling only Asian patients. Also, the patients enrolled in this trial were younger and had better renal function than in other studies [20]. Moreover, SEISMiC [19] only enrolled patients without signs or symptoms of hypoperfusion, which could have led to a selection bias by enrolling a relatively less sick cohort to determine the overall efficacy and safety of istaroxime. Finally, we included only three RCTs with a relatively small number of patients, thus impairing the generalizability of our findings.

## 5. Conclusions

Istaroxime improved hemodynamic and echocardiographic parameters, forming a promising strategy for AHF management. However, there were no significant differences in clinical outcomes, and there was an increase in adverse events with istaroxime. Furthermore, the current evidence is limited to a small number of RCTs; therefore, future large-scale phase III trials are needed to fully understand the short- and long-term effects of istaroxime on EC coupling and mitochondrial energetics, to further investigate hard cardiovascular outcomes, and to compare istaroxime with other inotropes.

## Figures and Tables

**Figure 1 diseases-11-00183-f001:**
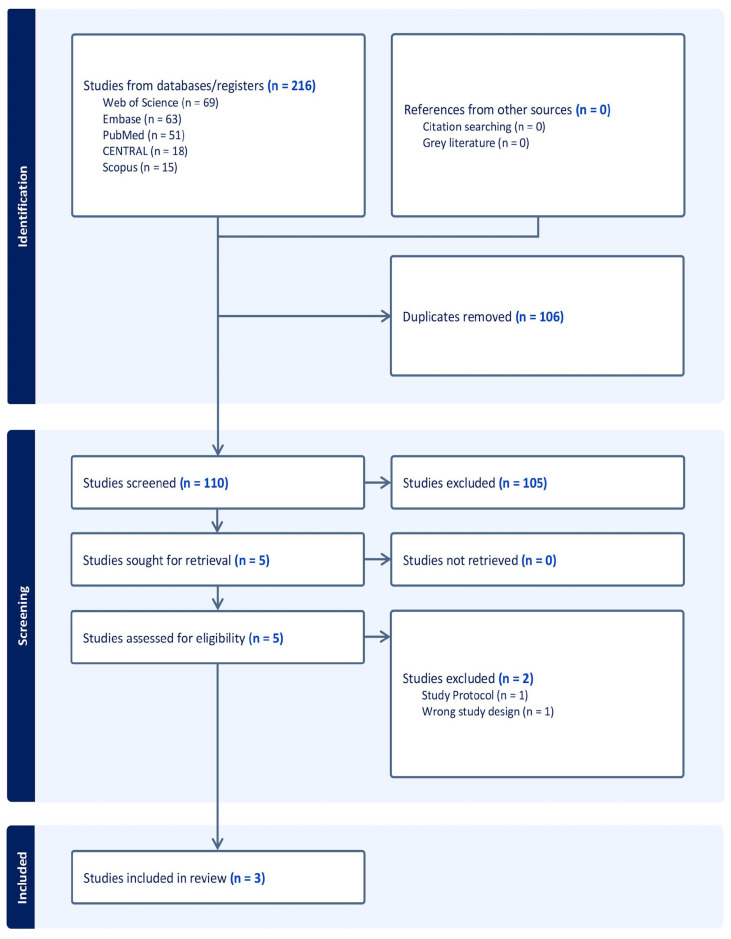
PRISMA flowchart of the screening process.

**Figure 2 diseases-11-00183-f002:**
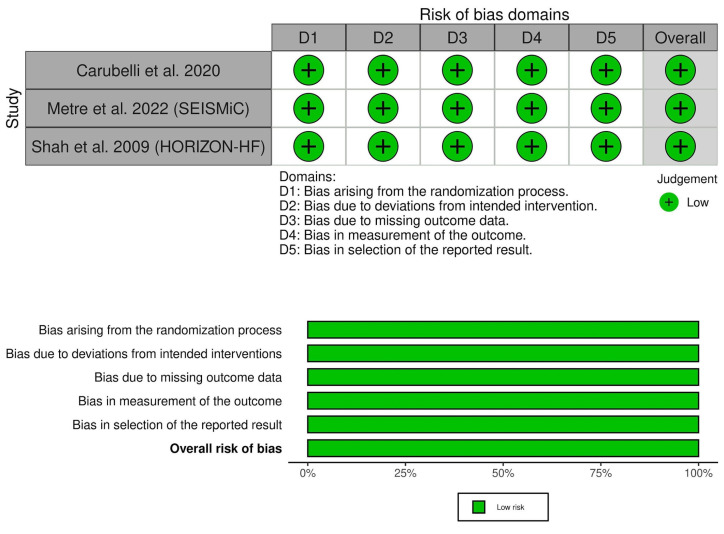
Quality assessment of the risk of bias in the included studies. Carubelli et al., 2020 [20], Metra et al., 2022 [19] (SEISMiC), Shah et al., 2009 [21] (HORIZON-HF).

**Figure 3 diseases-11-00183-f003:**
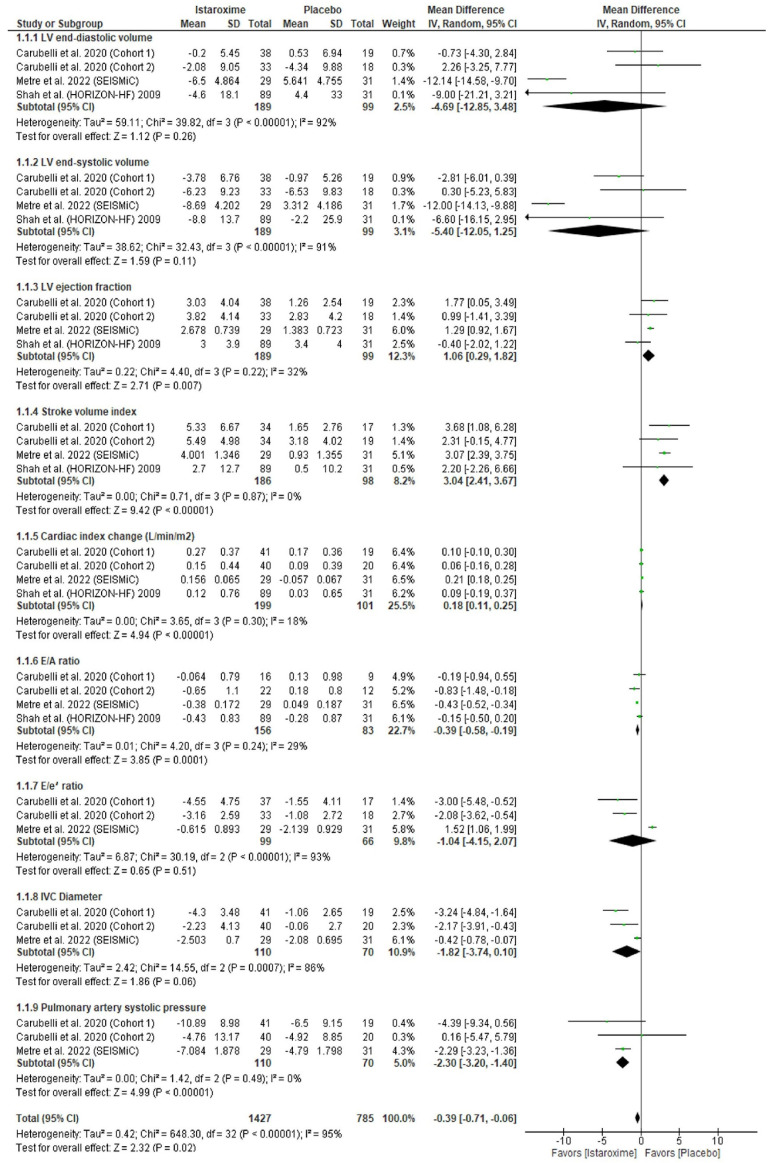
Forest plot of the echocardiographic parameters. CI: confidence interval; MD: mean difference. Carubelli et al., 2020 [20], Metra et al., 2022 [19] (SEISMiC), Shah et al., 2009 [21] (HORIZON-HF).

**Figure 4 diseases-11-00183-f004:**
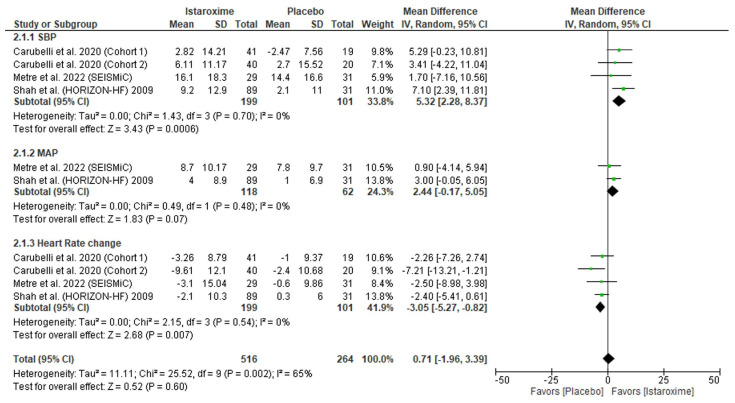
Forest plot of the hemodynamic parameters. CI: confidence interval; MD: mean difference. Carubelli et al., 2020 [20], Metra et al., 2022 [19] (SEISMiC), Shah et al., 2009 [21] (HORIZON-HF).

**Figure 5 diseases-11-00183-f005:**
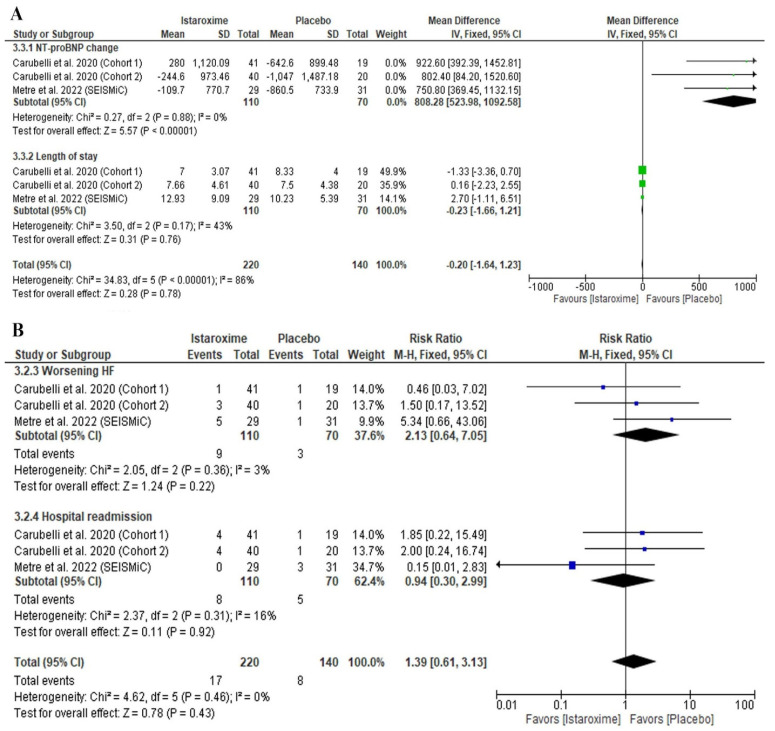
Forest plot of the clinical outcomes [(**A**) continuous outcomes; (**B**) dichotomous outcomes]. MD: mean difference, RR: risk ratio, CI: confidence interval. Carubelli et al., 2020 [20], Metra et al., 2022 [19] (SEISMiC), Shah et al., 2009 [21] (HORIZON-HF).

**Table 1 diseases-11-00183-t001:** Summary characteristics of the included RCTs.

Study ID	Study Design	Country	Total Participants	Istaroxime		Primary Outcome
				Dosage	Treatment Duration	
Carubelli et al., 2020 [20]	Double-blinded multicenter phase II RCT	Italy and China	120	0.5 μg/kg/min (cohort 1), 1.0 μg/kg/min in (cohort 2)	24 h	E/e′ ratio change
Metra et al., 2022 [19] (SEISMiC)	Double-blinded multicenter phase II RCT	US, Italy, Russia, Romania, and Poland	60	1.0–1.5 μg/kg/min	24 h	SBP change
Shah et al., 2009 [21] (HORIZON-HF)	Double-blinded multicenter RCT	US, Italy, Greece, Romania, and Poland	120	0.5, 1.0, or 1.5 μg/kg/min	6 h	Pulmonary capillary wedge pressure change

RCT: randomized controlled trial; SBP: systolic blood pressure; US: United States.

**Table 2 diseases-11-00183-t002:** Baseline characteristics of the participants.

Study ID	Number of Patients in Each Group		Age (Years) Mean (SD)		Gender (Male) N (%)		BMI Mean (SD)		SBP Mean (SD)		DBP Mean (SD)		HR Mean (SD)		Comorbidities N (%)											
	I	Pl	I	Pl	I	Pl	I	Pl	I	Pl	I	Pl	I	Pl	AF		HTN		PCI		CABG		DM		CKD	
															I	Pl	I	Pl	I	Pl	I	Pl	I	Pl	I	Pl
Carubelli et al., 2020 Cohort 1) [20]	41	19	60 (16)	58 (17)	34 (83)	16 (84)	25 (4)	25 (3)	105 (12)	105 (8)	.	.	72 (13)	77 (17)	8 (20)	5 (26)	17 (42)	11 (58)	.	.	.	.	14 (34)	5 (26)	12 (29)	5 (26)
Carubelli et al., 2020 Cohort 2) [20]	40	20	52 (13)	56 (16)	34 (85)	18 (90)	23 (4)	24 (4)	106 (10)	108 (10)	.	.	78 (11)	79 (13)	11 (28)	6 (30)	14 (35)	7 (35)	.	.	.	.	6 (15)	4 (20)	10 (24)	6 (30)
Metra et al., 2022 [19] (SEISMiC)	29	31	65 (10)	63 (13)	22 (76)	27 (87)	28 (6)	27 (6)	88 (3)	87 (3)	.	.	84 (16)	84 (19)	16 (55)	18 (58)	25 (86)	23 (74)	9 (31)	17 (55)	4 (14)	3 (10)	9 (31)	14 (45)	2 (7)	2 (6)
Shah et al., 2009 [21] (HORIZON-HF)	89	31	55 (11)	57 (10)	80 (90)	25 (81)	.	.	117 (12)	114 (15)	70 (7)	70 (8)	74 (9)	72 (11)	.	.	.	.	20 (22)	10 (32)	4 (4)	4 (13)	16 (18)	5 (16)	.	.

AF: atrial fibrillation; BMI: body mass index; CABG: coronary artery bypass graft; CKD: chronic kidney disease; DBP: diastolic blood pressure; DM: diabetes mellitus; HR: heart rate; HTN: hypertension; I: Istaroxime; ID: identification; N: number; Pl: placebo; PCI: percutaneous coronary intervention; SBP: systolic blood pressure; SD: standard deviation; . (period): missing data.

**Table 3 diseases-11-00183-t003:** GRADE evidence profile.

Certainty Assessment							Summary of Findings				
Participants (Studies) Follow-up	Risk of Bias	Inconsistency	Indirectness	Imprecision	Publication Bias	Overall Certainty of Evidence	Study Event Rates (%)		Relative Effect (95% CI)	Anticipated Absolute Effects	
							With [Placebo]	With [Istaroxime]		Risk with [Placebo]	Risk Difference with [Istaroxime]
**Echo-LV end-diastolic volume**											
288 (4 RCTs)	not serious	very serious ^a^	not serious	very serious ^b^	none	⨁◯◯◯ Very low	99	189	-	The mean echo-LV end-diastolic volume was **0**	MD **4.69 lower** (12.85 lower to 3.48 higher)
**Echo-LV end-systolic volume**											
288 (4 RCTs)	not serious	very serious ^a^	not serious	very serious ^b^	none	⨁◯◯◯ Very low	99	189	-	The mean echo-LV end-systolic volume was **0**	MD **5.4 lower** (12.05 lower to 1.25 higher)
**Echo-LV ejection fraction**											
288 (4 RCTs)	not serious	not serious	not serious	Serious ^b^	none	⨁⨁⨁◯ Moderate	99	189	-	The mean echo-LV ejection fraction was **0**	MD **1.06 higher** (0.29 higher to 1.82 higher)
**Echo-stroke volume index**											
284 (4 RCTs)	not serious	not serious	not serious	not serious	none	⨁⨁⨁⨁ High	98	186	-	The mean echo-stroke volume index was **0**	MD **3.04 higher** (2.41 higher to 3.67 higher)
**Echo-cardiac index change (L/min/m^2^)**											
300 (4 RCTs)	not serious	not serious	not serious	not serious	none	⨁⨁⨁⨁ High	101	199	-	The mean echo- cardiac index change (L/min/m^2^) was **0**	MD **0.18 higher** (0.11 higher to 0.25 higher)
**Echo-E/A ratio**											
239 (4 RCTs)	not serious	not serious	not serious	not serious	none	⨁⨁⨁⨁ High	83	156	-	The mean echo-E/A ratio was **0**	MD **0.39 lower** (0.58 lower to 0.19 lower)
**Echo-E/e**′ **ratio**											
165 (3 RCTs)	not serious	very serious ^a^	not serious	very serious ^b^	none	⨁◯◯◯ Very low	66	99	-	The mean echo-E/e′ ratio was **0**	MD **1.04 lower** (4.15 lower to 2.07 higher)
**Echo-IVC diameter**											
180 (3 RCTs)	not serious	very serious ^a^	not serious	very serious ^b^	none	⨁◯◯◯ Very low	70	110	-	The mean echo-IVC diameter was **0**	MD **1.82 lower** (3.74 lower to 0.1 higher)
**Echo-pulmonary artery systolic pressure**											
180 (3 RCTs)	not serious	not serious	not serious	Serious ^b^	none	⨁⨁⨁◯ Moderate	70	110	-	The mean echo- pulmonary artery systolic pressure was **0**	MD **2.3 lower** (3.2 lower to 1.4 lower)
**Hemodynamic-SBP**											
300 (4 RCTs)	not serious	not serious	not serious	Serious ^b^	none	⨁⨁⨁◯ Moderate	101	199	-	The mean hemodynamic outcomes-SBP was **0**	MD **5.32 higher** (2.28 higher to 8.37 higher)
**Hemodynamic-MAP**											
180 (2 RCTs)	not serious	not serious	not serious	Serious ^b^	none	⨁⨁⨁◯ Moderate	62	118	-	The mean hemodynamic outcomes-MAP was **0**	MD **2.44 higher** (0.17 lower to 5.05 higher)
**Hemodynamic-HR**											
300 (4 RCTs)	not serious	not serious	not serious	Serious ^b^	none	⨁⨁⨁◯ Moderate	101	199	-	The mean hemodynamic outcomes-HR change was **0**	MD **3.05 lower** (5.27 lower to 0.82 lower)
**Clinical-NT-proBNP change**											
180 (3 RCTs)	not serious	not serious	not serious	very serious ^b^	strong association	⨁⨁⨁◯ Moderate	70	110	-	The mean clinical outcomes-NT-proBNP change was **0**	MD **808.28 higher** (523.98 higher to 1092.58 higher)
**Clinical-length of hospital stay**											
180 (3 RCTs)	not serious	not serious	not serious	not serious	none	⨁⨁⨁⨁ High	70	110	-	The mean clinical outcomes-length of stay was **0**	MD **0.23 lower** (1.66 lower to 1.21 higher)
**Clinical-worsening HF**											
180 (3 RCTs)	not serious	not serious	not serious	very serious ^b^	none	⨁⨁◯◯ Low	3/70 (4.3%)	9/110 (8.2%)	**RR 2.13** (0.64 to 7.05)	43 per 1000	**48 more per 1000** (from 15 fewer to 259 more)
**Clinical-hospital readmission**											
180 (3 RCTs)	not serious	not serious	not serious	very serious ^b^	none	⨁⨁◯◯ Low	5/70 (7.1%)	8/110 (7.3%)	**RR 0.94** (0.30 to 2.99)	71 per 1000	**4 fewer per 1000** (from 50 fewer to 142 more)
**Safety-any adverse event**											
180 (2 RCTs)	not serious	not serious	not serious	very serious ^b^	none	⨁⨁◯◯ Low	48/70 (68.6%)	91/110 (82.7%)	**RR 1.26** (1.05 to 1.51)	686 per 1000	**178 more per 1000** (from 34 more to 350 more)
**Safety-any serious adverse event**											
180 (2 RCTs)	not serious	not serious	not serious	very serious ^b^	none	⨁⨁◯◯ Low	8/70 (11.4%)	14/110 (12.7%)	**RR 1.34** (0.58 to 3.11)	114 per 1000	**39 more per 1000** (from 48 fewer to 241 more)

CI: confidence interval; HF: heart failure; HR: heart rate; IVC: inferior vena cava; L: liter; LV: left ventricle; MAP: mean arterial pressure; MD: mean difference; RCT: randomized clinical trial; RR: risk ratio; SBP: systolic blood pressure. ^a^ I-square > 75%. ^b^ Wide confidence interval that does not exclude the risk of appreciable harm/benefit.

## Data Availability

All data are included in the manuscript.

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
