# Peer review of "Istaroxime for Patients with Acute Heart Failure: A Systematic Review and Meta-Analysis of Randomized Controlled Trials"

_diseases, 2023, doi:10.3390/diseases11040183_

Round 1

Reviewer 1 Report

Comments and Suggestions for Authors

The manuscript is interesting and updated since istaroxime is a very debated drug. I just have some minor points.

1. Check and correct some misspellings.

2. Add a graphical abstract with the main aim of the paper.

3. Improve the discussion section. Please find a manuscript that could help 

doi: 10.3390/jcm11247503.

doi: 10.4103/0976-500X.103705.

4. The limitations and the strengths should be better detailed.

Comments on the Quality of English Language

Ok

Reviewer 2 Report

Comments and Suggestions for Authors

- Firstly, this article is very similar to the one published very recently in the scientific journal Cureus on the same topic and entitled “Safety and Efficacy of Istaroxime in Patients With Acute Heart Failure: A Meta-Analysis of Randomized Controlled Trials”; Khalid Khan S, Rawat A, Khan Z, et al. (June 28, 2023) Safety and Efficacy of Istaroxime in Patients With Acute Heart Failure: A Meta-Analysis of Randomized Controlled Trials. Cureus 15(6): e41084. DOI 10.7759/cureus.4108.   - On the other hand, the article published in Cureus meta-analysis encompasses 211 patients with heart failure and includes a study published by Gheorghiade et al (Gheorghiade M, Blair JEA, Filippatos GS, et al., for the HORIZON-HF Investigators. Hemodynamic, echocardiographic, and neurohormonal effects of istaroxime, a novel intravenous inotropic and lusitropic agent: a randomized controlled trial in patients hospitalized with heart failure. J Am Coll Cardiol 2008; 51:2276 – 85.) and accordingly with editorial comment related to this topic “Istaroxime in Heart Failure New Hope…written by G. William Dec (Journal of the American College of Cardiology Vol. 51, No. 23, 200) “Gheorghiade et al. (12) describe the first randomized placebo-controlled trial of istaroxime in patients hospitalized with ADHF who underwent hemodynamic monitoring; ..”The study included patients aged 18 to 85 years with a left ventricular ejection fraction (LVEF) 35%, hospitalized with HF with a systolic blood pressure (SBP)50 and 90 mm Hg, heart rate (HR) 110 and 60 beats /min, and on standard HF therapy”.   -Why did not the authors Abuelazm et al, use data from the study by Gheorghiade et al []for publication in Diseases/MDPI) (n=30 Istaroxime 1 µg/Kg/min; Placebo, n=31)]?   Generally, the article is well written and meets the requirements of a Meta-Analysis study in accordance with was expressed in Lines 75-76: “the Preferred Reporting Items for Systematic Re-5 views and Meta-Analyses (PRISMA) guideline [21] and the Cochrane Handbook for Systematic Reviews and Meta-Analyses [22]”.     However, there are also some aspects that deserve changes that are referenced in red: (in red purposals to be potentially changed and also to be praised).   Abstract Line 32: wouldn't it be better to express the results in a different way and thus avoid the use of square brackets? From “(MD: 1.06, CI [0.29, 1.82], P = 0.007),” to (MD: 1.06 (95% CI: 0.29, 1.82; P = 0.007),” the same proposal for all results expressed in the abstract   Line 33: “(L/min/m2)” must be L/min/m2     Line 38: “provided some hematological and echocardiographic parameters”; change to: provided some hemodynamic and echocardiographic parameters,   Lines 86-91: in the article now under review we can read: "We included studies that followed the following PICO criteria: population (patients 86 with AHF), intervention (istaroxime irrespective of the dosage), comparison (placebo), outcomes { echocardiographic parameters [left ventricle (LV) end-diastolic volume, LV end-systolic volume, LV ejection fraction, stroke volume index, cardiac index, E/A ratio, E/e′ ratio, inferior vena cava (IVC) diameter, and pulmonary artery systolic pressure], hematological parameters [systolic blood pressure (SBP), mean arterial pressure (MAP), and heart rate (HR)], clinical outcomes { NT-proBNP change…”, However in the article of- Carubelli et al (2020) we can read “The primary efficacy endpoint was the change from baseline to 24 h after the start of the infusion in the E/e’ ratio assessed by tissue Doppler echocardiography. Secondary endpoints included the change from baseline to 24 h of other echocardiography parameters: LVEF, left ventricular end-systolic and end-diastolic volumes, stroke volume index, E, A, E/A ratio, and S′; changes…”. As can be confirmed in the article by Carubell et al, the parameter “pulmonary artery systolic pressure” is not included, which is supposed to..as well , in the study by Metra et al, the inferior vena cava diameter parameter was not analysed.   - Lines 93-94: it reads “… safety outcomes [the incidence of any adverse events and any serious adverse events]}, but there is no description in the text regarding serious results/side effects (for example: newly diagnosed coronary artery disease , cardiac arrest, ventricular tachycardia, ventricular fibrillation...) and not serious (for example cardiovascular and/or gastrointestinal... which side effects (injection site pain…nausea, vomiting); because this topic appears in the discussion there should be a paragraph about the side effects found (Istaroxime and Placebo).   - In table 1 there is a small error: the name of the author of the second study included is Metra and not Metre; (change in all tables) and in the text where Meter is; -  In table 2, columns AF, HTN, PCI, CABG, DM, CKD are missing the percentage (%); Table 3 “deformatting with words in the columns cut to another line (e.g. comparisi   on);         3.4. Echocardiographic Parameters: incorrectly used as a caption for this Table and not as a subtitle for the following topic   In the document sent, the line count starts again from 3.3. Risk of Bias and Certainty of Evidence from line 150… becomes 1; so in line 8 ); 3.4. Echocardiographic Parameters: incorrectly used as a caption for this Table and not as a sub-title for the following topic);     - In line 9: it reads “95% CI [0.29, 1.82], P = 0.007); but it should be read: 95% CI (0.29, 1.82; P = 0.007) and thereafter standardize.   - Line 13: pulmonary artery systolic pressure (MD: -2.30 1..: units missing mmHg; regarding LV end-diastolic volume: fmissing mL; …end-systolic volume …mL; IVC …missing mm.   Line 27: where it says “3.5. Hematological Parameters”; should read 3.5. Hemodynamic Parameters;   Line 28: “Istaroxime was significantly associated with increased SBP”: mmHg missing Line 29: “decreased HR”: missing bpm…MAP…mmHg Line 34: “Istaroxime was significantly associated with increased NT-proBNP level; increase or decrease and units? Line 36: “hospital length of stay”: the units …days   In Figure 4, the data relating to NT-Pro-BNP from the Carublelli study cohort cannot be understood in the column Mean Difference IV, fixed, 95%: why the result of 802.4? The final values are also not understood (does it favor the placebo? What does this mean??)   Lines 44-48: It reads “Istaroxime was significantly associated with increased incidence of any adverse  events rate (RR: 1.26 with a 95% CI [1.05, 1.51], P = 0.01). However, there was no difference between istaroxime and placebo regarding any serious adverse event (RR: 1.34 with a 95% 46 CI [0.58, 3.11], P = 0.49) (Figure S1). Our results were homogenous in the incidence of any  adverse events (P = 0.34, I2 = 0%) and any serious adverse event (P = 0.52, I2 = 0%)”; but what adverse events? Serious and non-serious…comparing with placebo.     - Line 54 : hematological?…hemodynamic   - Line 74: it reads “..reticulum Ca+2 adenosine triphosphatase isoform 2a (SERCA2a) [6,13,14].”; but in the quote [6} there is no reference to istaroxime   - Lines 79 and 85: I'm not sure if references 29 and 31 (“…homeostasis role in cardiomyocyte contractility has been extensively investigated [29…e 31 “(Line 85: “mechanism of NCX, can result in slowed relaxation [31 ].contain the content described in the discussion   Lines 142-143: “To the best of our knowledge, this is the first meta-analysis to investigate the efficacy and safety of istaroxime for AHF management, constituting the gold-standard evidence to date.”; unfortunately it is not the first meta-analysis to be published (see Safety and Efficacy of Istaroxime in Patients With Acute Heart Failure: A Meta-Analysis of Randomized Controlled Trials. Cureus 15(6): e41084. DOI 10.7759/cureus.4108.How can you read in this Cureus article: Review began06/21/2023; Review ended 06/25/2023; Published 06/28/2023. It is possible to admit that there are some differences (studies included, sample size...) and that the current article has been written in the same period… (june/July 2023?...but…   Line 160: “Istaroxime improved blood pressure, tachycardia, and some echocardiographic”; it should read Istaroxime increase blood pressure, reduced tachycardia, and some echocardiographic…   Line 183: “BMI Basal metabolic index”; Body Mass Index  

As already mentioned, this article is well written and meets the requirements of a Meta-Analysis study. The different chapters are correctly prepared from a scientific point of view (Introduction, methods, results...). The discussion is very well built and according to the results obtained. The limitations are well delineated and the conclusions agreed on the basis of the results obtained. The references are apparently few but very current and well integrated with the theme under analysis (correlated with few clinical publication about Istaroxime in the treatment of Acute Heart Failure). From a formal point of view, it respects the norms required by the Diseases/MDPI Scientific Publication. And the main conclusion is unfortunately correct: “Furthermore, the current evidence is limited to a small number of RCTs; therefore, future large scale phase III trials are needed to fully understand the short and long-term effects of istaroxime”.  

Comments on the Quality of English Language

No

Reviewer 3 Report

Comments and Suggestions for Authors

Interesting article

I would be more immediate in describing the drug by saying that it contains the combined pharmacological benefits of digitalis and levosimendan

Limitations of the meta-analysis population: young population

Isn't the average heart rate a bit slow in a population with acute heart failure?

About the heart rate modulation that is reduced with Istaroxime, briefly mention the use of Landiolol (https://doi.org/10.1093/eurheartjsupp/suac121.143). Surely there will be more qualified references than the one I have proposed.

Cite this study on Levosimendan. Tumminello G J Cardiovasc Dev Dis. 2021 Oct 7;8(10):129. doi: 10.3390/jcdd8100129. PMID: 34677198; PMCID: PMC8539734. It is a meta-analysis like yours and has somewhat of the same methodological limitations and can help you enrich the discussion

The style of the article also needs to be more appealing

Evaluate the opportunity of a summary graph with the pros and cons of Istaroxime

You still did a good job, just needed some tidying up and some minor revisions
